# Phytochemical, Morphological, and Physiological Variation in Different Ajowan (*Trachyspermum ammi* L.) Populations as Affected by Salt Stress, Genotype × Year Interaction and Pollination System

**DOI:** 10.3390/ijms241310438

**Published:** 2023-06-21

**Authors:** Gita Mirniyam, Mehdi Rahimmalek, Ahmad Arzani, Parisa Yavari, Mohammad R. Sabzalian, Mohammad Hossein Ehtemam, Antoni Szumny

**Affiliations:** 1Department of Agronomy and Plant Breeding, College of Agriculture, Isfahan University of Technology, Isfahan 8415683111, Iran; gmirniyam@ymail.com (G.M.); a_arzani@cc.iut.ac.ir (A.A.); parisa.y.373@gmail.com (P.Y.); sabzalian@cc.iut.ac.ir (M.R.S.); hehtemam@cc.iut.ac.ir (M.H.E.); 2Department of Horticulture, College of Agriculture, Isfahan University of Technology, Isfahan 8415683111, Iran; 3Department of Food Chemistry and Biocatalysis, Wrocław University of Environmental and Life Sciences, 50-375 Wroclaw, Poland

**Keywords:** ajowan, essential oil composition, bioactive compounds, salt stress, inbreeding, pollination

## Abstract

In the present research, 28 populations of ajowan (*Trachyspermum ammi* L.) were evaluated for agro-morphological traits and essential oil yield in two consecutive years. Then, selected ajowan populations from these two years were used for further morphophysiological and biochemical studies under different salinity levels (control, 60, 90, and 120 mM NaCl). The main components of the oil were thymol (32.7–54.29%), γ-terpinene (21.71–32.81%), and *p*-cymene (18.74–26.16%). Salt stress caused an increase in essential oil content in the Esfahfo and Qazvin populations. The highest total phenolic and flavonoid contents were found in the Arak population grown in 60 mM NaCl (183.83 mg TAE g^−1^ DW) and the Yazd population grown in 90 mM NaCl (5.94 mg QE g^−1^ DW). Moreover, the Yazd population exhibited the strongest antioxidant activity based on DPPH (IC_50_ = 1566 µg/mL) under 60 mM NaCl and the highest reducing power (0.69 nm) under 120 mM NaCl. The results revealed that low and moderate salt stress improves the phytochemicals of ajowan seeds, which are useful for pharmaceutical and food applications. In this research, some morphological traits, as well as essential oil yield, were evaluated in open pollinated versus self-pollinated plants. As a result, plant height, number of flowering branches, and crown diameter significantly decreased in some populations, while a significant increase was obtained for number of flowers per umbel and seed numbers per umbel. Finally, self-pollination of ajowan might provide new insights for further breeding programs to increase oil or thymol content in ajowan.

## 1. Introduction

We are now witnessing a growing interest in replacing synthetic aromatic compounds with natural counterparts. Distributed in semi-arid and arid parts of the world [1], ajowan, or sprague, (*Trachyspermum ammi* L.), is a medicinal and industrial plant of the Apiaceae family with white flowers and small brownish fruits. As a good source of secondary metabolites, ajowan seeds have been used in food and pharmaceutical applications [2]. Ajowan seeds exhibit a variety of health properties, which include antimicrobial, antioxidant, nematicidal, anti-inflammatory, carminative, and sedative effects [2]. Thymol is considered a valuable phenolic monoterpene that is widely used in food products for its high antioxidant and antimicrobial capacities [3]. Interestingly, ajowan seeds have also been found to be a good source of thymol. The seeds are commonly used to preserve and fortify certain food products due to their resistance to processing and their health-promoting compounds [4].

Phenolic compounds are the most important secondary metabolites of the plant kingdom, with numerous biochemical and molecular functions, such as defense, signaling, mediating auxin transport, scavenging free radicals, and antioxidant activity [5]. Polyphenols including flavonoids have several benefits for human health, such as scavenging free radicals, regeneration, and protection of other dietary antioxidants. The aforementioned antimicrobial and anti-inflammatory properties of *T. ammi*, like those of other medicinal herbs, are attributed to its phenolic constituents. Nevertheless, the ajowan seed is not only putatively beneficial to human health, but it is also a valuable source of amino acids, dietary fiber, vitamins, and minerals [6,7].

Until now, a few studies have been conducted on the variations in the essential oil content of ajowan populations [1,2,4], while others have concentrated on the polyphenols in and antioxidant activity of oils [2]. Moreover, the effects of salt stress have been explored in some medicinal plants, including *Myrthus communis* [8], *Mentha canadensis* [9], *Thymus vulgaris* [10], and *Salvia mirzayanii* [11]; nonetheless, there is scant information available for ajowan about the effects of salt stress on its physiological, essential oil, and biochemical attributes.

The reproductive system of medicinal plants has provided new insights for the genetic improvement of secondary metabolites [12]. In this regard, studying the pollination type and fertility is critical for their conservation and improvement [13]. Moreover, self-pollinated plant production plays a crucial role for improving medicinal plants to help fix some of the morphological and phytochemical traits [14]. In this regard, the effect of self-pollination on different traits has been reported in some other plants, such as alpine eryngo (*Eryngium alpinum* L.) [15], wild parsnip (*Trachymene incise* Rudge.) [16], (*Foeniculum vulgare* Mill.) [17], and (*Mentha* spp.) [18].

Pollination, which is essential for the process of fertilization and production of fruits and/or seeds, is one of the limiting factors for ajowan productivity that significantly contributes to its productivity. In nature, only about 5 percent of the flowers are self-pollinated, and 95 percent require another source, such as an animal pollinated system. Insect pollination accounts for approximately 90 percent of total pollination of crops [19]. Finally, knowledge about pollination systems is fundamental for further breeding programs in this species [20]; however, there are no reports regarding the effects of pollination systems on morphological and yield-related traits in ajowan.

Therefore, the present study was conducted: (1) to evaluate the morphological, oil, and yield-related traits in 28 ajowan populations grown in two consecutive years; (2) to evaluate the effects of salt stress on the physiological traits, seed yield and related traits, essential oil composition, polyphenolic contents, and antioxidant capacity of selected ajowan populations; (3) to assess the effect of self-pollination on some important morphological traits; and (4) to perform multivariate analyses for better classification and interpretation of data.

## 2. Results

### 2.1. Two-Year Study of Morphological, Oil, and Yield-Related Traits of 28 Populations

The results of a two-year analysis of variance revealed significant differences among the studied genotypes for all traits. Additionally, the effect of the year was significant for most the studied genotypes. The interaction of G × Y (genotype × year) was also significant for some traits that were studied (Table 1).

Morphological traits also revealed a height variation in the studied population. The height ranged from 63.66 to 100.5 cm (Table 2). The tallest population was Khormo, while the shortest one was Farsfars. The number of flowering branches ranged from 6.16 (Ardebil) to 20.83 (Khormo). The Arak and Farsfars populations recorded the smallest inflorescence diameter, while the IPK2 and Khorsar populations possessed the largest inflorescence diameter. The number of umbels varied considerably from 29.83 to 273.39 (Table 2). IPK3 and Farsfars possessed the highest and the lowest number of umbellules per inflorescence, respectively. The number of flowers per umbel ranged from 114 (Farsfars) to 282.67 (Yazshah). The largest and smallest numbers of seeds per umbel were recorded for Yazshah (565.33) and Farsfars (228), respectively. The smallest (278.6 cm^2^) and largest (1548.7 cm^2^) crown cover belonged to Khorbi and Arakkho populations, respectively. The two Yazsad and Khormo populations exhibited the highest values for 1000 seed weight and seed yield per plant (0.96 g and 72.45, respectively), while Ardebil (0.67 g) and Tehran (9.85 g) recorded the lowest values.

One of the most important traits for selection in the ajowan population was the yield of essential oils. A high variation in the oil yield in the first year was observed. Accordingly, the highest and the lowest essential oil yields were recorded for the Yazd (5.51) and Farsfars (1.20) populations, respectively. An increase in oil content was observed in most of the populations studied in the second year. The highest essential oil yield was observed in Yazd (5.58), while the lowest was from Yazshah (1.74). Similar ranges were also reported for Indian ajowan populations [21]. High variation in the two studied years could have resulted from environmental fluctuations, as previously reported in other Apiaceae plants, including ajowan [22] and fennel [23]. Similarly, an elevation was obtained for most of the morphological traits in the second year (Table 2). All morphological traits revealed a large increase compared to the previous year.

#### Hierarchical Cluster Analysis

The clustering patterns of the 28 ajowan (*Trachyspermum ammi* L.) populations based on their morphological data and the oil yield obtained from the Ward method for the two years are presented in Figure 1 and Figure 2, respectively.

In the first year, group 1 included the Khorsar, Arakkho, Yazsad, Yazd, Esfahfo, Farsmar, and Ardebil populations with the highest essential oil yield, while group 2 classified 16 populations with a moderate to high range of 1000 seed weight. Five populations, including Yazshah, Esfahgh, IPK3, Hamadan, and Khormo, were placed in group 3 and were characterized mostly by a high plant height.

In the second year, group 1 included Khorsar, Arakkho, and Yazsad; and group 2 consisted of 11 populations. Group 3 included the Ardebil, Hamedan, Esfahgh, Araksha, and Khorsa populations with a high plant height. Group 4 consisted of four populations with mostly moderate 1000 seed weights, while group 5 was characterized by a high essential oil yield (Figure 2).

The results of the two-year analysis revealed large variation with respect to the most of the traits. Therefore, the essential oil yield and 1000 seed weight were used to select four superior populations for the second experiment. Consequently, two populations that showed a medium quantity (Esfahfo and Qazvin) and two (Arak and Yazd) that revealed a large number of the mentioned traits were chosen for the second experiment (Table 2).

### 2.2. Effects of Salt Stress on Morphophysiological, and Phytochemical Traits

#### 2.2.1. Essential Oil Content

Based on the results obtained, the oil content of the *T. ammi* populations was found to be considerably influenced by salt stress treatments, while the essential oil yield ranged from 2.16 to 4.77% under the control condition. The highest and lowest yields were recorded for Arak and Qazvin, respectively, both of which exhibited strongly reduced levels under the examined NaCl concentrations (Table 3). However, the Esfahfo and Qazvin populations recorded an increase in their essential oil (EO) yields.

At low salt stress (LS), the highest (4.39%) oil yield was recorded for Yazd, while Arak exhibited the lowest (3.01%) yield. A similar trend was observed under the moderate stress (MS) condition such that the Arak and Esfahfo populations recorded the highest (4.22%) and lowest (2.64%) oil yields, respectively. Under severe stress (SS) conditions, Qazvin recorded the highest essential oil yield, while the lowest (3.77%) belonged to Esfahfo (Table 3).

Previous studies reported different ranges of essential oil yield for different ajowan populations collected from different countries. They reported a range of 2% to 4% for the essential oil yield extracted from ajowan seeds [24]. These results were confirmed by [25], who also reported a range of 2–4.4% for the essential oil yield extracted by hydro-distillation from some Iranian ajowan populations, whereas higher ranges of 2.5% to 6.1% have also been reported for EO yields from Iranian ajowan populations [26]. In the present experiment, salinity stress was observed to decrease the essential oil yield in the Yazd and Arak populations. This finding could be due to the additional energy demand by plant tissues as a result of the lower availability of carbon concentrations during the growth stage that resulted in reduced oil accumulation [9]. Furthermore, the increased production of volatile compounds in Esfahfo and Qazvin under elevated salt stress could be attributed to the elevations of oil gland density in these populations [27].

#### 2.2.2. Essential Oil Composition

Table 4 reports all the EO constituents detected in the four studied populations under the different salt treatments. Clearly, there are high chemical polymorphisms among the Iranian ajowan populations with thymol, γ-terpinene, and *p*-cymene identified as the main components. It is seen in this table that the amount of thymol ranged from 32.7 in Qazvin under the LS treatment to 54.29 in Qazvin under the control conditions. The highest (32.81) and lowest (21.71) amounts of γ-terpinene belonged to Qazvin (LS) and Esfahfo (SS), respectively. *p*-cymene content varied from 26.16 in Esfahfo (LS) to 18.74 in Arak (C). Among the few studies reporting on the composition of essential oils of Iranian ajowan populations, they identified γ-terpinene (48.07%), *p*-cymene (33.73%), and thymol (17.41%) as the major constituents of one population grown in Firoozabad, Iran [28]. In the Esfahan population, the most abundant components of the oil were reportedly thymol (44.5%), γ-terpinene (26.6%), *p*-cymene (21.6%), limonene (1.1%), and carvacrol (0.3%) [29].

The effect of salt stress on thymol content has also been reported in the *Thymus* genus [10]. Whereas the populations examined in the present study showed different trends in thymol accumulation in their seeds, the differences observed between thyme and ajowan plants with respect to their thymol content might be attributed to their harvested organs. While it is the seeds of the ajowan species that are harvested for their high thymol content, the edible leaves of *Thymus* are harvested for thymol extraction.

#### 2.2.3. Yield-Related Traits

According to the results obtained, the number of seeds per plant of the *T. ammi* populations was found to be considerably influenced by salt stress treatments. Salt stress caused a significant reduction in the seed yield per plant of *T. ammi* (Table 3). A maximum reduction in seed yield was observed with LS treatment. The number of seeds per plant varied from 47.66 to 68.75 g under the control condition, and the lowest and highest were recorded for Arak and Yazd, respectively (Table 3). Under low salt stress (LS), the largest (67.34) number of seeds per plant was recorded for Esfahfo while Arak exhibited the smallest (40.46) number of seeds. Under moderate stress (MS) treatment, the Esfahfo and Qazvin populations recorded the highest (49.50) and lowest (29.22) numbers of seeds, respectively. Under severe stress (SS) conditions, Arak recorded the largest (34.10) number of seeds per plant, while the smallest number (28.59) came from Qazvin (Table 3). 

NaCl in moderate and severe stress levels in growth medium caused a marked reduction in the number of seeds per plant in *T. ammi* [29]. Such an adverse effect of salinity on growth and seed yield was observed earlier in a number of crops, e.g., alfalfa [30,31], carrots [32], and cumin [33].

#### 2.2.4. Total Phenolic and Flavonoid Contents

The studied ajowan populations also showed different trends with respect to their accumulation of phenolic compounds. All the samples exhibited high TPC values ranging from 61.76 in Yazd (C) to 183.83 mg TAE g^−1^ DW in Arak (LS), followed by Qazvin (LS) (157.32 mg TAE g^−1^ DW). The accumulation of phenolic compounds in each plant is the result of such varied parameters as phenological stage, extraction process, agricultural application, and storage conditions [34]. In plants exposed to abiotic stresses, the rate of cellular oxidative damage can be controlled by the plant’s capacity to produce antioxidants [35]. However, the accumulation of phenolic compounds may be different in different plants as a result of salinity stress [36]. For instance, phenolic compounds were shown to decrease in broccoli [37] in response to salt stress, whereas NaCl treatment elevated TPC levels in maize [38] and red peppers [39].

Similarly, the studied ajowan populations exhibited substantial differences with regard to their flavonoid content. While, Yazd (MS) recorded the highest TFC content (5.94 mg QE g^−1^DW), Yazd (C) exhibited the lowest (3.48 mg QE g^−1^ DW). Additionally, a significant increase in TFC was observed under moderate salinity stress (MS), but higher salt concentrations were observed to cause decreases in the levels of TFC (Table 3).

#### 2.2.5. DPPH Radical Scavenging

Comparisons were performed to detect variations in the scavenging of DPPH free radicals in the ajowan extracts (Figure 3). The IC_50_ values were found to vary from 1566.985 µg/mL to 5889.99 µg/mL. More specifically, the extract of the Yazd population subjected to MS and SS showed the strongest antioxidant activities (1566.985 and 1657.46 µg/mL, respectively), while those from Qazvin (C) and Esfahfo (C) demonstrated the weakest activities (5889.99 and 5671.98 µg/mL, respectively). The variation in IC_50_ observed among different species might be interpreted with recourse to the diversity in their polyphenolic components [40]. It likely can be suggested that plants activate metabolite biosynthesis as part of a complex antioxidant defense mechanism when they are exposed to salt stress and that the production of phenolic compounds might be part of an alternative strategy adopted by plants to respond to stressful conditions. The antioxidant capacity of *Thymus* species has been well researched. The most relevant chemotypes of *Thymus* species have been reported to be rich in phenolic monoterpenes, such as thymol and carvacrol [41]. In most such studies, phenolics, due to their chemical structures, which allow them to donate hydrogen to free radicals, were introduced as the main factor contributing to the antioxidant activity of the species [42]. Moreover, [43] reported that, based on the observations, the highest antioxidant capacity recorded might be due to large amounts of phenolic components of the species. Studying the effect of salt stress on the antioxidant activity of Apiaceae plants, [21] reported a large variation in the antioxidant activity of some Indian Apiaceae spices based on their results from DPPH assay. Similarly, [44] used the DPPH method to observe a large variation in the antioxidant capacity of three *Ferula* species.

#### 2.2.6. Reducing Power

The reducing power of the studied ajowan populations was found to rise with increasing EO concentrations (Figure 4). Clearly, the greatest absorbance antioxidant capacity at 700 nm was obtained from Yazd (SS) (0.84) in 500 mg/L, while Arak (C) exhibited lower activity than BHT (butylated hydroxytoluene). From among the ajowan populations, in absorbance at 700 nm, Yazd (SS), Yazd (MS), Esfahfo (SS), and Yazd (LS), recorded reducing powers of 0.84, 0.69, 0.44, 0.39, and 0.39, respectively, which were higher than those recorded for the other populations (Figure 4). Similarly, [45] reported that the strongest reducing power was observed in *A. pachycephala* at concentrations of 300 and 500 mg/L. In a similar study based on the reducing power model, [8] found that myrtle subjected to salt stress showed an increased antioxidant activity with increasing extract concentrations.

#### 2.2.7. Cluster and Principal Component (PCA) Analysis

Cluster analysis was performed using the main essential oil components, TPC and TFC to detect any similarities among the studied ajowan populations. The results are illustrated in the dendrogram in Figure 5. The analysis provided further information regarding the distribution of ajowan populations in terms of their essential oil yields, and it suggested that diversified chemical compositions resulted from the different salt stress conditions investigated. The obtained results grouped the ajowan populations into the following four clusters. The first group consisted of the two accessions of Qazvin (C) and Arak (LS), both of which were rich in thymol (54.29, 50.05). The second group was divided into two subgroups. The first consisted of Qazvin (MS), Arak (MS), Yazd (LS), and Esfahfo (MS), the main components of which were TFC (5.35, 4.37) and TPC (147, 125.99). Finally, Yazd (MS) and Esfahfo (SS), with higher quantities of *p*-cymene (20.55, 19.21), were assigned to the second subgroup. The third group consisted of Esfahfo (C), Arak (SS), Yazd (SS), and Arak (C). The main components of this group were large quantities of essential oil yield (4.77, 3.22). The fourth group consisted of Qazvin (LS), Qazvin (SS), Yazd (C), and Esfahfo (LS), the main components of which were γ-terpinene (32.81, 22.75) (Figure 6). Based on the structural similarity of thymol and carvacrol, it may be suggested that the rate of their conversion into each other may be affected by environmental factors such as salt stress [46].

PCA was also performed to group the investigated populations in terms of their main oil components and the other metabolites studied. The PCA result revealed that the first and second components explained 69.32% of all the variations observed (Table 5), while the first component explained 44.40% of the total variation. Finally, PC2 explained 24.91% of the total variance. The first PC (PC1) had a positive correlation with *p*-cymene (0.480) and γ-terpinene (0.533) but a negative correlation with thymol (−0.580). It should be noted that thymol is an isomer of carvacrol, while *p*-cymene is considered a precursor to both compounds [47]. Finally, PC2 showed a large positive contribution by EO content (0.405) and thymol (0.220) but a negative correlation with phenol (−0.690). Comparison of cluster and PCA results showed similar trends in most cases, such as Yazd (SS), Esfahfo (C), Arak (C), and Arak (SS), which possessed large amounts of essential oil yield. Likewise, Qazvin populations in control conditions were rich in thymol. The four populations of Qazvin (SS), Yazd (C), Esfahfo (LS), and Qazvin (LS) formed a single group characterized by higher levels of *p*-cymene and γ-terpinene. The seven populations of Qazvin (SS), Arak (SS), Arak (MS), Yazd (LS), Esfahfo (MS), Esfahfo (LS), and Arak (LS) formed a single group characterized by higher TPC. Overall, the studied ajowan populations exposed to different salt level concentrations and control conditions were successfully distinguished based on their phytochemical traits and main EO components.

#### 2.2.8. Correlations among the Components

Correlation analysis demonstrated the relationships between all the measured traits and the main essential oil components under different conditions. According to Table 6, in the control condition, negative correlations were recorded between thymol and γ-terpinene (−0.961 *) and between DPPH and γ-terpinene (−0.990 **). Under low salt stress (LS), a positive correlation was observed between DPPH and *p*-cymene (0.947 *). The result of the correlation analysis under moderate stress (MS) conditions showed negative correlations between thymol and γ-terpinene (−0.994 **), while the correlation between total phenolic and thymol (0.970 *) and between reducing power and total phenolic content (0.969 *) were positive. Finally, in a severe stress (SS) environment, a positive correlation was shown between γ-terpinene and *p*-cymene (0.944 *) and between EO content and *p*-cymene (Table 7). γ-terpinene is the major precursor to the biosynthesis of thymol, and *p*-cymene is considered a by-product of this pathway. Previous reports have shown different trends in the accumulation of these components in thyme leaves [43].

Since ajowan seeds were used in the present study for essential oil analysis, the discrepancies observed between the results obtained for thyme and ajowan could be explained with regard to the organs in which the oils accumulated. This explanation is confirmed by the results reported elsewhere that highlighted changes in monoterpene frequencies are based on phenological differences and harvested organs [4,48]. Finally, the positive correlations between the precursors and final products might be attributed to the complete transformation of the precursors.

#### 2.2.9. Physiological Traits

##### Malondialdehyde (MDA) and Hydrogen Peroxide (H_2_O_2_)

For malondialdehyde (MDA), the results of analysis of variance revealed that the main effect of populations and salinity and the interaction effect of salinity on populations were significant (Table 8). The effects of populations × salinity for MDA showed that the largest and smallest amounts were related to the Qazvin and Esfahfo populations in control conditions with 5.36 and 1.46 nmol/m of leaf fresh weight, respectively (Table 3). It has been reported that leaf MDA content at 6 dS/m NaCl level has significantly increased in different *Sesamum indicum* cultivars compared with controls [49]. Unsaturated fatty acids are the main constituents of membrane lipids that are prone to peroxidation by free radicals due to salinity stress [50]. In this regard, MDA content is indicative of oxidative damage [51].

A significant difference in the content of hydrogen peroxide (H_2_O_2_) was observed among the populations in the control environment and under salinity stress treatments. H_2_O_2_ was decreased at salinity levels of 6 dS/m. The results of population × salinity showed that the highest and lowest values of this trait were related to Qazvin populations under control conditions with 2.40 mmol/g and Arak at a stress level of 6 dS/m with 0.45 mmol/g (Table 3).

##### Antioxidant Enzymes Activity

The results of the statistical analysis showed that there is a significant difference among populations, different salinity levels, and the interaction effect of salinity on populations influencing the activity of guaiacol peroxidase and ascorbate peroxidase (Table 8). Interaction effects of salinity on populations for guaiacol peroxidase enzyme revealed that the highest and lowest values of this trait belong to Qazvin populations at a stress level of 12 dS/m with 0.277 FW U mg^−1^ and to the Isfahanfo population under 12 dS/m level with 0.012 FW U mg^−1^ (Table 3). Additionally, the largest amount of ascorbate peroxidase enzyme was related to Arak populations in control and Arak conditions at a stress level of 6 dS/m with 0.025 FW U mg^−1^. The lowest was related to Yazd populations at a stress level of 6 dS/m and Qazvin at a stress level of 9 dS/m (Table 3).

According to ANOVA, the main effects of populations, salinity, and the interaction of salinity on populations were significant for chlorophyll a and chlorophyll b (Table 8). The interaction effects of salinity in chlorophyll a revealed that the highest and lowest rates of this trait were related to Esfahfo populations at a stress level of 9 dS/m with 0.36 mg/g and Yazd populations at a stress level of 6 dS/m with 0.05 mg/g (Table 3). Additionally, the interaction effects of salinity on populations for chlorophyll b showed that the highest and lowest values of this trait were related to Isfahfo populations at a stress level of 9 dS/m with 0.09 mg/g and Yazd at a stress level of 9 dS/m (0.023 mg/g) (Table 3). Due to the direct role of chlorophyll a in photosynthesis and dry matter production, this trait can also be effective in increasing this difference. Most of the previous reports indicated that the chlorophyll content decreases under salinity stress, and the old and necrotic leaves begin to fall as the salinity period continues. Decreases in chlorophyll content as a result of salinity stress have also been reported for cotton [52], pumpkin [53], and spinach [54].

##### Carotenoids

ANOVA results showed that the main effects of populations, salinity, and the interaction of salinity on populations were significant for the carotenoid trait (Table 8). The effects of salinity interaction for carotenoids showed that the highest and lowest amounts were obtained in Esfahfo populations at a stress level of 9 dS/m with 0.155 mg/g and Yazd at a stress level of 9 dS/m with 0.03 mg/g (Table 3).

#### 2.2.10. MANOVA Analysis

Two-way multivariate analysis of variance for each of the stress factors and genotypes by three specified methods showed that the measured traits were able to differentiate between genotypes and stress levels. The results are illustrated in Table 9.

### 2.3. Morphological Traits in Self- and Open-Pollinated Populations

The mean values of morphological traits are presented in Table 10. Qazvin possessed the highest plant height among all populations (122.67 cm), viz. Open pollination-derived populations (117.67 cm), whereas the lowest height was found among open-pollinated plants (100 cm) in Qazvin in a self-pollination-derived population (104.33 cm) in Arak.

Populations showed different responses to the pollination system for number of flowering branches between the open- and self-pollinated Qazvin and Arak populations. Additionally, in the Yazd population, there was no significant difference in inflorescence diameter between open- (2.67 cm) and self-pollinated plants. In the other population, crown cover diameter significantly increased as a result of open pollination (1227.22 cm^2^) compared with the self-pollination (596.60 cm^2^) in Esfahfo. In contrast, in the Qazvin population, the number of umbels was significantly decreased after open pollination (40.33) compared with self-pollinated plants (65.33). Some trends were also obtained in *Foeniculum vulgare* Mill. [17]. These differences in reaction to pollination systems among the populations could be due to other factors affecting morphological traits in medicinal plants, such as genetic factors and evolutionary histories of populations [55]. Moreover, quantitative variation in traits may be attributed to environmental factors, such as temperature, relative humidity, irradiance, and photoperiod, and also to genetic factors, such as the pollination system, as evident in this study. Inbreeding depressions (IDs) for the number of umbellules per inflorescence, number of flowers per umbel, and number of seeds per umbel are illustrated in Table 10.

The essential oil yield of open-pollinated plants was greater than that with self-pollination [56]. The essential oil yield in the open-pollinated population of Yazd was 5.51%, while in the self-pollinated population, it was 4.54%. In Esfahfo, the essential oil yield in the open-pollinated population was 5.32%, while in its self-pollinated population, it was 1.50%. In this regard, open-pollinated populations in Qazvin and Arak possessed a higher essential oil yield than the corresponding self-pollinated populations [18]. These results may indicate that genetic recombination after self-pollination and particularly after open-pollination might lead to changes in the genetic background of ajowan regarding essential oil yield.

## 3. Discussion

In the present research, thymol was the major compound in ajowan seed. Thymol is an aromatic and oxygenated monoterpene [57,58]. Furthermore, monoterpene accumulation can be highly affected, not only by the phenological stages but also by harvesting time [25]. In the case of ajowan, seeds are harvested at full maturity, and monoterpenes mostly begin to increase from the full flowering stage to seed maturity [2]. Moreover, salt stress reportedly affects the biosynthesis of isoprenoids as a result of its influence on isoprene subunits. The different trends observed in thymol accumulation in the populations examined in the present study make it difficult to draw definitive conclusions about its quantities in different populations. However, comparison of the control and severe stress treatments revealed that Arak, Esfahfo, and Qazvin showed decreases in their thymol content with the severe stress treatment. Several explanations have been suggested for the decrease in thymol content. One explanation claims that the dissipation energy mechanism involved in isoprenoid changes under stress conditions is responsible, as the changes are attributed to the subunits available for the biosynthesis of isoprenoids or related compounds [59]. In the present study, plants subjected to severe stress (SS) were used the available carbon sources for the production of carbohydrates that are necessary for grain filling [46,60]. Finally, the radical scavenging mechanism has also been suggested regarding changes in metabolites during stress conditions [26]. Regardless of the explanation, certain compounds with high antioxidant activities might be involved to address free radicals.

In the present study, the changes in polyphenolics were also investigated. Plants have been reported to employ different mechanisms for distributing flavonoids among their subcellular sections. Metabolically, plant polyphenols, such as flavonoids and phenolics, are biosynthesized through several pathways [59]. The underlying mechanism involved in flavonoid functions is based on the chelating or chipping process. Some reports have evidenced the enhancement of phenolics in various plant structures and organ systems under salinity stress condition [35]. Moderate salinity stress induces the normal saline tolerance pathway through increasing flavonoid contents [39]. Hence, the variations observed in the studied ajowan populations, as well as the different salt stress conditions, might have led to the increase in the polymorphisms in flavonoids and their accumulation.

Physiological aspects of salt stress in ajowan populations were also investigated and discussed. Hydrogen peroxide (H_2_O_2_) is an active signal molecule, and its accumulation leads to a wide range of plant responses to environmental stresses, as these reactions are interdependent. Increasing the level of environmental stresses increases the production of reactive oxygen species (ROS), such as hydrogen peroxide, leading to increased damage to plant cells [61]. In the present study, the selected ajowan populations revealed a high physiological variation in reactions to salt stress. Previous studies have also highlighted that different species and populations can reveal different reactions to stress and release various types of antioxidants that neutralize the effects of signal molecules and increase plant tolerance to stress [62]. Salinity tolerant cultivars have less hydrogen peroxide than sensitive cultivars [63]. Therefore, the hydrogen peroxide content under stress conditions can be used as a suitable indicator for selection of tolerance to salinity [64]. This kind of variation was also observed in Apiaceae plants, including *Carum carvi* L. [65] and *Foeniculum vulgare* Mill [66].

## 4. Materials and Methods

### 4.1. Plant Materials and Growing Conditions

Twenty-five populations of ajowan originating from Iran and three populations originating from the Leibniz Institute of Plant Genetics and Crop Plant Research, Gatersleben, Germany (IPK), were used in this study (Table 11). The Iranian samples were identified by Dr. Mozaffarian using Flora Iranica, and their herbarium specimens were deposited for research at the Institute of Forests and Rangelands, Islamic Republic of Iran (Table 11). The seeds were sown in a randomized complete block design (RCBD) under field conditions at Lavark Research Farm of Isfahan University of Technology on 2 March 2017 and 2018 (Appendix A). The research farm is located at 32° North and 51° East. The average annual rainfall and minimum and maximum absolute temperatures are 122.8 mm, 9.1 °C, and 23.4 °C, respectively. The soil texture of the field was a clay loam with a bulk density of 1.4 g cm^–3^ and a pH of 7.8. The plot size was 1 × 2 m^2^ and the individual plants were spaced 30 cm apart. The seeds were harvested on 5 September 2017 and 2018.

### 4.2. Salt Stress Treatments

Four selected ajowan populations, namely Yazd, Esfahfo, Qazvin, and Arak, were chosen based on their geographical origin, seed yield, and oil yield (experiment I data) and were used in this study. A pot experiment was performed under greenhouse conditions with an average temperature of 25 °C and average humidity of 50%. Each pot contained 9 kg of soil at a soil to sand ratio of 3:1 (Appendix A). After the complete establishment of the plants in the pot and before flowering, the treatments were applied from the lowest levels to the highest, and the duration of stress for each level was 30 days.

A factorial experiment with a randomized complete block design layout replicated three times was used. Four treatments of 0 (control), 60 (low stress = LS), 90 (moderate stress = MS), and 120 mM NaCl (severe stress = SS) were conducted. The salt solution was gradually applied at the initial irrigation. At the end of the experiment, the saturated soil electrical conductivity (ECe) of the pots was determined using an EC meter. The recorded ECe values were as follows: control: 3.1 dS m^−1^; LS: 4.8 dS m^−1^; MS: 6 dS m^−1^; and SS: 8.5 dS m^−1^. After the complete application of stresses during the growth period, the following traits were evaluated.

### 4.3. Essential Oil Extraction

A Clevenger-type device was applied to extract the oil from the ajowan seeds. For this purpose, 50 g of seeds were subjected to instrument using 500 mL of distilled water, and the boiling time was set as 6 h. Finally, the oil yield was reported based on dry matter [4].

### 4.4. GC/MS Analysis

Essential oil components were identified using the Agilent 7890 gas chromatography with mass selective detector (Agilent Technologies, Palo Alto, CA, USA) equipped with a HP-5MS column (30 m × 0.25 mm, 0.25 μm film thickness). The oven temperature was 60 °C for 4 min before it was elevated to 260 °C at a rate of 4 °C/min. The temperature of the GC injector port was 290 °C, and it was 300 °C for the detector. Helium was applied as the carrier gas at a flow rate of 2 mL/min. The mass unit was set at an ion source temperature of 240 °C and ionization voltage of 70 eV. The retention indices of the oils were calculated using the retention time of the n-alkane series (C_5_–C_24_) [67], as well as using NIST. The amounts of oils were obtained from the GC/MS peak.

### 4.5. Flavonoid and Phenolic Evaluation

To prepare a sample extract, 100 mL of 80% methanol was added to 6 g of the seed samples and shaken gently for 24 h. Then, the solution was filtered to remove the solid residues and collected for further experiments. For TFC evaluation, the aluminum chloride colorimetric method was applied based on the procedure in [68]. Initially, a volume of 125 µL of the extract was added to 75 µL of a 5% NaNO_2_ solution. The studied samples were kept in the dark for 6 min before a solution of 10% Alcl_3_ (150 μL) was added to each and maintained in the dark for an additional 5 min to complete the reaction. The absorbance of ajowan samples was determined at 510 nm. Finally, the TFC was presented in milligrams of quercetin equivalents (QE) per gram of the extract.

Phenolic compounds were determined based on the Folin–Ciocalteu method [43]. For this purpose, 2.5 mL of the Foline–Ciocalteu reagent (1:10 diluted with distilled water) was mixed with 0.5 mL of the methanolic extract. The samples were incubated for 5 min at room temperature, and then 2 mL of 7.5% sodium carbonate solution was added in a tube test. The mixture was maintained at 45 °C in a hot water bath for 15 min. Then, the absorbance of the mixture was calculated at 765 nm using a spectrophotometer. Tannic acid equivalents (TAEs) were used as the reference standard, and the total phenolic content (TPC) was expressed as milligrams of TAEs per gram of each extract on a dry basis.

### 4.6. Antioxidant Capacity

#### 4.6.1. DPPH Assay

Antioxidant capacity was determined using the DPPH (2, 2-diphenylpicrylhydrazyl) method, as described in [69] with some modifications. For this purpose, ajowan extracts were prepared in the different concentrations of 50, 300, and 500 mg/L in methanol. A control (A_B_ Control) including methanol and DPPH solution was also prepared. Different concentrations of butylated hdroxytoluene (BHT), used as a positive standard, were added to 1 mL of the methanol solution (0.2 mM) of the DPPH reagent, and absorbance was measured at 517 nm against the blank. When readings were augmented, the inhibition percentage of the samples was calculated according to the following formula:$$\%\mathrm{inhibition}=\frac{A_{B-A_{A}}}{A_{B}}\times 100$$
where A_A_ and A_B_ are the absorbance values of the DPPH radical in the presence of the plant extract sample and the control, respectively. The inhibition percentage was plotted versus the sample concentration, and 50% of the inhibitory concentration (IC_50_) of the DPPH values was defined by linear regression analysis.

#### 4.6.2. Reducing Power

Antioxidant capacity was also evaluated based on the reducing power of each extract as described by [70]. For this purpose, 2.5 mL of the methanol extracts (with the different concentrations of 50, 300, and 500 mg/L) were mixed in a phosphate-buffered solution (2.5 mL, 0.2 M, pH = 6.6) with 2.5 mL of 1% potassium ferricyanide [K_3_Fe (CN) _6_]. Then, 2.5 mL of trichloroacetic acid (10%) was added, and the reaction mixture was centrifuged at 3000 rpm for 10 min. Finally, the supernatant obtained after centrifugation (2.5 mL) was mixed with 2.5 mL of deionized water and 0.5 mL of ferric chloride (0.5 mL, 0.1%). The absorbance was obtained at 700 nm versus a blank.

### 4.7. Malondialdehyde (MDA) Content

At the end of stress, 1 g of fresh leaf sample was mixed with 5 mL of 0.1% TCA. Then, the homogenized mixture was centrifuged at 10,000 rpm for 5 min. Subsequently, 500 μL of the supernatant was removed, and 2 mL of 20% TCA + 0.5% TBA solution was added. The samples were heated at 95 °C for 30 min. After final centrifuge at 10,000 rpm for 15 min, the absorbance of the supernatant was read at 532 nm [71].

### 4.8. Hydrogen Peroxide (H_2_O_2_) Content

For this evaluation, the samples were powdered with liquid nitrogen, and then 5 mL of 0.1% TCA was gradually added. Next, the homogenized mixture was centrifuged, and 500 μL of potassium phosphate buffer (prepared from KH_2_PO_4_ and K_2_HPO_4_) was added. Finally, the absorbance was measured at 390 nm [72].

### 4.9. Ascorbate Peroxidase (APX) Activity

To measure the activity of ascorbate peroxidase, 250 mM phosphate buffer, 1.2 mM hydrogen peroxide, 0.5 mM ascorbic acid, and 0.1 mM EDTA were mixed. The reduction in absorption due to ascorbic acid peroxidation at 290 nm for 2 min was measured. Absorption changes per minute were applied to calculate enzyme activity [73].

### 4.10. Guaiacol Peroxidase (GPX) Activity

The reaction consisted of 25 mM potassium phosphate buffer, 40 mM hydrogen peroxide, and 20 mM guaiacol mixed with 100 μL of enzyme. Increased adsorption was recorded by tetraglycol formation at 470 nm for 3 min [74].

### 4.11. Protein Assay

This method was used to measure the protein concentrations of total plant extracts according to [75]. For this purpose, the fresh plant materials were applied for this experiment. Two milliliters of 0.1 M phosphate buffer was mixed with fresh plants, and the samples were then centrifuged at 15,000 rpm for 12 min at 4 °C. The absorption was measured using a spectrophotometer at a wavelength of 595 nm.

### 4.12. Measurement of Chlorophyll Content

After applying salinity stress, the amount of chlorophyll in the leaves was evaluated by the method in [76]. 

### 4.13. Self-Pollination Studies

In 2018, plants from each populations were divided in two parts, of which the half umbels were bagged from the start of inflorescence emergence until seed harvesting for obligatory self-pollination, whereas the other half were left uncovered for open, pollination. At the end of the summer, seeds were separately harvested from the self- and open-pollinated umbels at the full maturity stage, and finally, traits such as plant height, number of flowering branches, crown cover diameter, number of umbels, number of seeds per umbel, and essential oil yield were measured (Appendix A).

### 4.14. Statistical Analysis

Analysis of variance for traits was performed in the form of a randomized complete block design in three replications using SPSS software, version 20, SAS software, version 9.4, and STATGRAPHICS software, ver. XVII. Means were compared using the least significant difference (LSD) method. To group populations, Ward’s cluster analysis was used based on the square of the Euclidean distance using STATGRAPHICS software, version 16.2.04, and to verify the results of the cluster analysis, PCA analysis was used by STATGRAPHICS software, version 16.2.04. MANOVA was performed to examine the effects of the experimental treatments from a multivariate perspective by SAS software, version 9.4.

## 5. Conclusions

In the present study, a two-year morphological study was performed to assess the variation in ajowan populations for two years in terms of morphological and oil yield. Moreover, phytochemical and physiological responses to salt stress were evaluated in selected populations. In the third experiment, morphological traits were also evaluated for different pollination-derived plants, viz. self- and open-pollinated ones (S and OP). Accordingly, high genotypic variation was found among and within the S and OP populations for most of the measured traits, emphasizing the high potential for genetic analysis of the studied traits. Furthermore, Arak (control) > Yazd (control) > Yazd (LS) > Qazvin (SS) recorded superior EO yields, suggesting that they are the best populations for producing the largest amounts of favorable metabolites under non-stress, low stress, and severe salt stress conditions. The major EO compounds were identified as thymol, *p*-cymene, and γ-terpinene. In the present research, the Yazd and Esfahfo populations revealed promising efficiency in terms of yield-related traits, oil and thymol content, and antioxidant capacities. Finally, the results of the present study provide new insights for further breeding programs in ajowan.

## Figures and Tables

**Figure 1 ijms-24-10438-f001:**
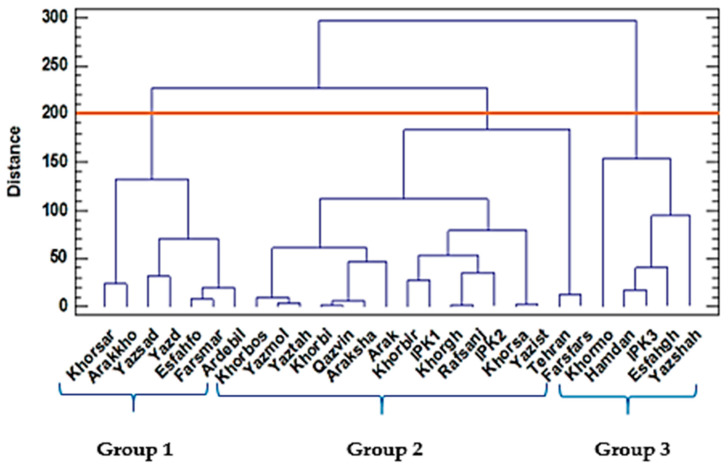
Dendrogram of 28 ajowan (*Trachyspermum ammi* L.) populations based on morphological traits and essential oil content using the Ward method, based on the squared Euclidean dissimilarity calculated in 2017.

**Figure 2 ijms-24-10438-f002:**
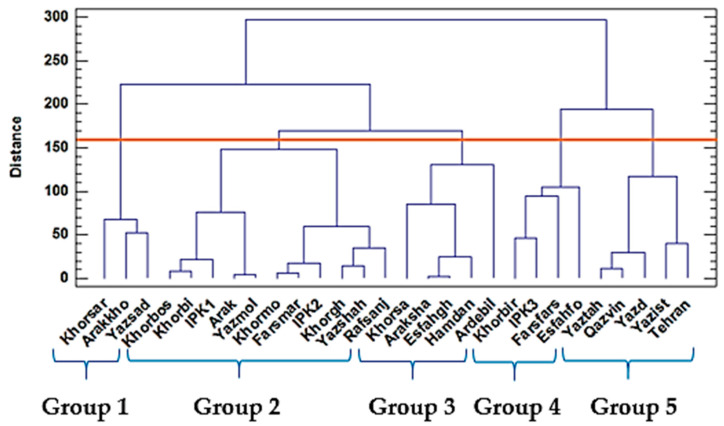
Dendrogram of 28 ajowan (*Trachyspermum ammi* L.) populations based on morphological traits and essential oil content using the Ward method, based on the squared Euclidean dissimilarity calculated in 2018.

**Figure 3 ijms-24-10438-f003:**
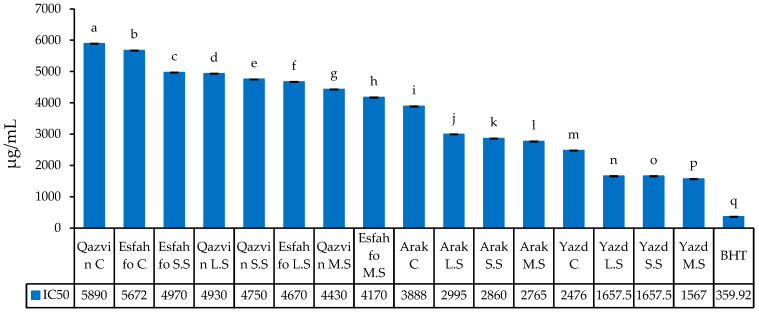
IC_50_ (µg/mL) of ajowan populations extracts compared to BHT. BHT: butylated hydroxytoluene. C: control; LS: low stress; MS: moderate stress; SS: severe stress; Means with a different letter are statistically significant at the 5% level of probability.

**Figure 4 ijms-24-10438-f004:**
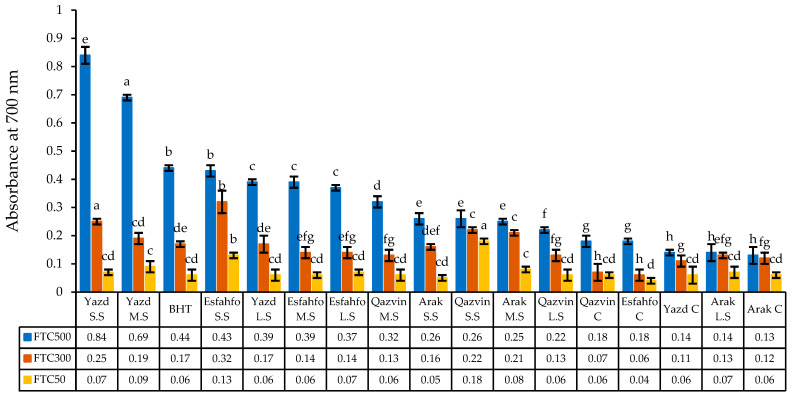
Evaluation of antioxidant activity based on reducing power in ajowan seed extracts compared to BHT. Control (C), 60 (low stress = LS), 90 (moderate stress = MS), and 120 dSm-1 (severe stress = SS). BHT: butylated hydroxytoluene. Means with a different letter are statistically significant at the 5% level of probability.

**Figure 5 ijms-24-10438-f005:**
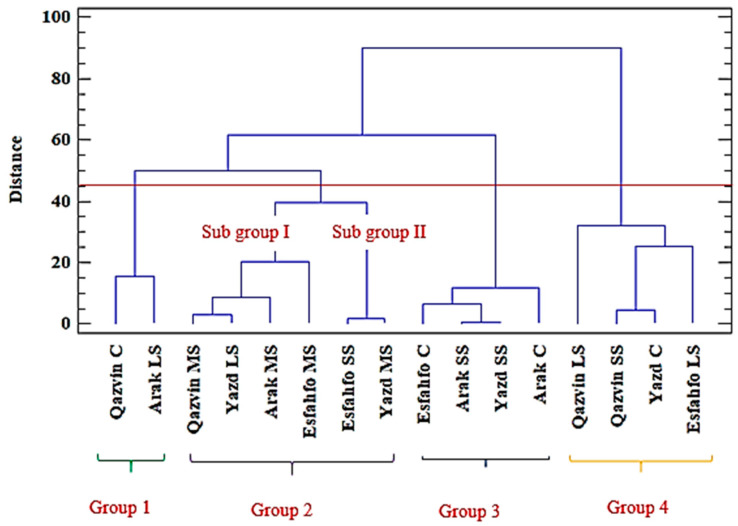
Dendrogram of four ajowan populations under different salinity levels using the Ward clustering method. C: Control; LS: Low stress; MS: Moderate stress; SS: Severe stress.

**Figure 6 ijms-24-10438-f006:**
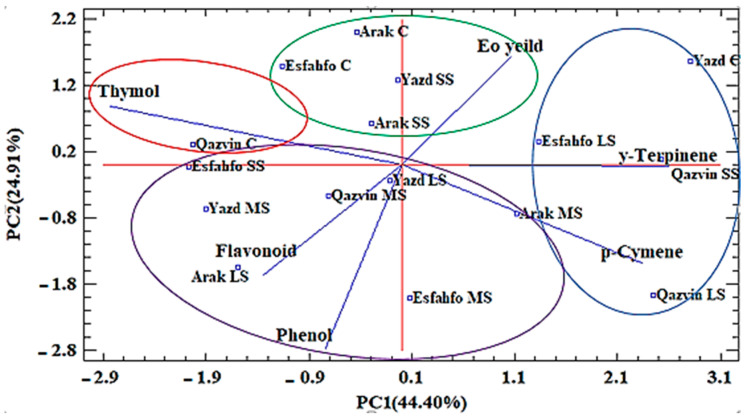
Principle component analysis of ajowan populations based on major essential oil components and other phytochemical traits. C: Control; LS: Low stress; MS: Moderate stress; SS: Severe stress.

**Table 1 ijms-24-10438-t001:** Analysis of variance for morphological traits and essential oil content of 28 ajowan (*Trachyspermum ammi* L.) populations.

Means of Square												
Source of Variation	df	Plant Height	No. of Flowering Branches	Inflorescence Diameter	No. of Umbels	No. of Umbellule per Inflorescence	No. of Flowers per Umbel	No. of Seeds per Umbel	Crown Cover Diameter	One Thousand Seed Weight	Seed Yield per Plant	Essential Oil Yield
Year (Y)	1	24396.01 **	2648.14 **	60.84 **	785321.11 **	1618.82 **	840793.21 **	336,3172 ^ns^	357,643.06 **	0.0370 **	386,539.88 **	1.040 **
Rep (year)	4	195.21	23.70	0.40	9048.53	5.86	3063.36	12,253.47	228,663.37	0.0006	4356.98	0.045
G	27	381.72 **	110.35 **	0.80 **	25,217.35 **	16.58 **	21,926.23 **	87,704.90 **	596,830.26 **	0.0500 **	10,394.04 **	6.360 **
Y × G	27	323.70 **	130.90 **	0.92 **	24389.70 **	23.23 **	23,900.90 **	95,603.80 **	4386.02 ^ns^	0.0001 ^ns^	10,673.01 **	0.036 ^ns^
Residual	108	99.51	8.62	0.31	2260.86	4.39	3868.95	15475	36513	0.0003	1517.52	0.0531

^ns^: non-significant. **: Significant at 1% level of probability.

**Table 2 ijms-24-10438-t002:** Means of morphological characteristics and essential oil content of 28 ajowan (*Trachyspermum ammi* L.) populations in two years (2017 and 2018).

Genotype	Height (cm)		No. of Flowering Branches		Inflorescence Diameter		No. of Umbels		No. of Umbellule per Inflorescence		No. of Flowers per Umbel		No. of Seeds per Umbel		Crown Cover Diameter (cm^2^)		One Thousand Seed Weight (g)		Seed Yield per Plant		Essential Oil Yield (%)	
	2017	2018	2017	2018	2017	2018	2017	2018	2017	2018	2017	2018	2017	2018	2017	2018	2017	2018	2017	2018	2017	2018
1	87	109.83	14.5	18.1	3.91	5.5	58	105.2	13	18.5	219.83	337.5	439.67	675	1462.7	1896.9	0.72	0.75	31.93	80.31	4.35	4.51
2	84.1	107.83	14.16	11.6	3.16	3.7	76	262.7	11.83	17.1	172.67	164.5	345.33	329	351.5	569.8	0.87	0.92	52.48	86.37	4.35	4.47
3	91.6	115.83	12.66	43	3	4.2	78.67	389.3	10.833	21	139.5	471.5	279	943	741.2	1056	0.76	0.79	27.81	286.8	3.63	4.13
4	77.1	139.67	14.66	18.3	2.83	4.5	102.83	199	13.16	20	180.5	232.8	361	465.7	449.1	697.1	0.58	0.61	21.43	61.94	3.01	3.16
5	77.6	101.33	12.66	15	2.66	4.5	82.5	167	13.16	15	198.33	205.3	396.67	410.7	278.6	489.1	0.8	0.82	41.31	58.44	4.36	4.43
6	100.5	104.5	20.83	26.1	2.75	4.1	273.39	135	13	19.5	197.83	349.1	395.67	698.3	688	987.9	0.72	0.75	72.45	71.39	4.09	4.16
7	88.33	107.33	12.5	15.6	3.41	4.4	140	162.8	13.16	15.8	192.67	280	385.33	560	778.7	1097.4	0.7	0.73	32.09	67.61	2.79	2.9
8	87.33	108.67	10.5	16.8	3.5	4.1	57.5	127.6	13.66	20.6	219.83	264	439.67	528	1548.7	1906.2	0.92	0.96	43.38	63.6	5.23	5.2
9	76	129.83	10	10.5	2.16	4.1	65.5	147.3	12	18.8	137.5	275.6	275	551.3	563.9	833.2	0.83	0.86	15.53	71.9	5.08	5.13
10	67.83	116	11.16	14.5	2.65	4	88.33	342	12.83	18.5	204.33	324	408.67	648	491.9	755.7	0.77	0.8	34.7	187.27	4.61	4.66
11	95.16	109.5	14	10.5	3.38	4.3	143	413.3	16	18.5	273.17	362.1	546.33	724.3	458.3	705.6	0.73	0.75	58.53	228.05	2.78	2.98
12	85.66	112.66	12.33	12.5	3.13	3.8	132.5	173.3	12	20.3	180.67	275.1	361.33	550.3	669.6	969.5	0.91	0.94	39.11	91.9	4.61	4.69
13	85.16	104.66	14	25.5	2.86	5.2	110.83	204.1	12	22.6	172.83	360.5	345.67	721	626.4	905.1	0.85	0.87	32.19	128.71	4.03	4.15
14	74.5	105.33	10	28	3.08	4.8	81.33	194.5	15.16	15.6	193.67	273.3	387.33	546.7	771.1	1059.9	0.96	0.98	20.46	106.39	4.72	4.85
15	84	91.5	7.66	19.5	3.91	3.8	61.5	154.8	13.83	16.3	282.67	236	565.33	472	606.7	886.2	0.63	0.66	22.19	51.56	1.62	1.74
16	72.66	107.16	12.16	11.5	2.83	4.5	130.25	250.5	11.83	23.8	171.17	470	342.33	940	537	832.8	0.58	0.61	23.46	127.19	3.43	3.53
17	75.5	111.83	7.83	13.1	3.25	4.8	29.83	235.8	15.83	20.6	253.67	404.8	507.33	809.7	354.4	584.9	0.92	0.96	14.16	177.6	5.51	5.58
18	103.33	118.83	17.66	15.3	3.75	3.7	145.65	372.1	13.5	16.6	263.33	211.3	526.67	422.7	357.9	585.5	0.7	0.72	69.33	114.77	4.23	4.37
19	75.5	100	10.5	20.5	3.4	4.1	86	326.8	13.83	29.6	217.5	614	435	1228	694.9	1021.4	0.74	0.77	24.11	315.02	5.32	5.52
20	77.33	97.83	10.83	25.5	2.91	5.1	102.33	188.1	14.33	22.3	192	510.1	384	1020.3	417.5	672	0.78	0.8	25.71	159.55	4.73	4.94
21	76.5	95.83	8	18.5	2.25	5.5	56.33	80.6	11.5	20.5	125.5	444.3	251	888.7	644.8	954.6	0.71	0.74	9.85	53.59	2.07	2.28
22	66.16	112.66	6.16	42.6	3.25	4.1	43.5	550.3	13.83	20.1	205.67	238.8	411.33	477.7	625.3	896.1	0.67	0.7	12.13	171.82	3.8	3.91
23	83.16	104.5	8.66	15.3	3.25	3.8	44.09	134.5	14	16.5	238.83	397.6	477.67	795.3	358.7	612.5	0.75	0.78	20.09	74.06	4.34	4.4
24	63.66	88.5	7.66	27.6	2.33	3.9	61	399.8	9.33	24.6	114	493.3	228	986.7	432.9	848.8	0.7	0.73	10.61	290.74	1.2	2
25	87.66	93.33	13	21.6	3.08	3.3	163.67	344	12	17	175.67	302.8	351.33	605.7	755.2	1098.7	0.72	0.75	40.72	118.15	3.53	3.67
26	98	104.66	10.16	12.5	3.33	3.3	60.67	76	14.16	15.5	136.5	164.6	273	329.3	434.8	733.9	0.71	0.76	11.6	19.3	4.04	4.18
27	85.5	87.83	11.66	22.3	4.08	4.6	103.25	111.6	13.16	16.1	187	291.1	374	582.3	560.4	839.4	0.73	0.76	23.89	46.9	4.33	4.44
28	98.66	114.66	15.16	21	3.66	4.6	99.75	258.1	16.16	20.8	243.67	497.3	487.33	994.7	932.1	1289.8	0.74	0.75	41.16	247.64	4.17	4.39
LSD	14.72	17.79	4.76	4.8	1.08	0.6	22.38	107.7	3.72	3.1	93.47	109.5	186.94	219	288.2	335.5	0.03	0.03	33.72	83.64	0.38	0.36

**Table 3 ijms-24-10438-t003:** Contents of total phenolics, total flavonoids, and essential oil yields of four selected *Trachyspermum ammi* populations.

Species	TFC (mg QE g^−1^ DW)	TPC (mg TAE g^−1^ DW)	ESO (%)	No. of Seeds per Plant	MAD	H_2_O_2_	GPX	APX	Pro	Chla	Chlb	Car
Qazvin (C)	3.50 ^k^	124.32 ^h^	2.16 ^o^	48.98 ^g^	5.36 ^a^	2.40 ^a^	0.22 ^a^	0.007 ^d^	0.23 ^ab^	0.07 ^c^	0.031 ^c^	0.041 ^d^
Qazvin (LS)	4.22 ^g^	157.32 ^b^	3.11 ^l^	52.55 ^e^	0.86 ^d^	1.94 ^b^	0.067 ^b^	0.02 ^b^	0.23 ^a^	0.18 ^a^	0.06 ^a^	0.08 ^b^
Qazvin (MS)	4.37 ^f^	135.15 ^d^	3.53 ^j^	29.22 ^o^	1.81 ^d^	1.59 ^c^	0.075 ^b^	0.006 ^c^	0.23 ^a^	0.22 ^b^	0.063 ^b^	0.104 ^b^
Qazvin (SS)	3.49 ^k^	119.26 ^i^	4.26 ^d^	28.59^p^	4.35 ^a^	1.95 ^a^	0.277 ^a^	0.014 ^a^	0.238 ^a^	0.136 ^b^	0.040 ^a^	0.076 ^b^
Esfahfo (C)	3.99 ^h^	64.85 ^m^	3.22 ^k^	63.79 ^c^	1.46 ^d^	2.20 ^b^	0.11 ^c^	0.016 ^b^	0.24 ^a^	0.17 ^b^	0.037 ^b c^	0.077 ^c^
Esfahfo (LS)	3.76 ^j^	77.32 ^l^	3.74 ^i^	67.34 ^b^	3.62 ^b^	2.12 ^a^	0.261 ^a^	0.01 ^c^	0.24 ^a^	0.13 ^b^	0.04 ^a^	0.07 ^c^
Esfahfo (MS)	5.35 ^b^	127.44 ^f^	2.64 ^n^	49.50 ^f^	3.10 ^c^	1.97 ^a^	0.014 ^c^	0.011 ^a^	0.24 ^a^	0.36 ^a^	0.089 ^a^	0.155 ^a^
Esfahfo (SS)	5.00 ^c^	133.58 ^e^	3.77 ^h^	38.01 ^l^	2.28 ^d^	1.52 ^d^	0.012 ^d^	0.010 ^b^	0.243 ^a^	0.179 ^a^	0.040 ^a^	0.092 ^a^
Arak (C)	4.26 ^g^	81.86 ^k^	4.77 ^a^	47.66 ^h^	2.51 ^b^	0.91 ^d^	0.18 ^b^	0.025 ^a^	0.23 ^b^	0.19 ^b^	0.065 ^ab^	0.099 ^b^
Arak (LS)	4.42 ^f^	183.83 ^a^	3.01 ^m^	40.46 ^,j^	3.70 ^a^	0.45 ^d^	0.022 ^c^	0.025 ^a^	0.23 ^a^	0.23 ^a^	0.05 ^a^	0.10 ^a^
Arak (MS)	4.71 ^d^	125.99 ^g^	4.22 ^e^	39.94 ^k^	3.84 ^a^	1.54 ^c^	0.089 ^a^	0.010 ^b^	0.24 ^a^	0.14 ^c^	0.070 ^ab^	0.083 ^b^
Arak (SS)	4.40 ^f^	99.13 ^j^	4.10 ^f^	34.10 ^m^	3.30 ^c^	1.73 ^b^	0.208 ^b^	0.014 ^a^	0.240 ^a^	0.177 ^a^	0.048 ^a^	0.078 ^b^
Yazd (C)	3.48 ^k^	61.76 ^n^	4.61 ^b^	68.75 ^a^	1.69 ^c^	1.66 ^c^	0.17 ^b^	0.014 ^c^	0.23 ^b^	0.34 ^a^	0.080 ^a^	0.154 ^a^
Yazd (LS)	4.50 ^e^	147.00 ^c^	4.39 ^c^	59.13 ^d^	2.81 ^c^	1.41 ^c^	0.078 ^b^	0.002 ^d^	0.23 ^a^	0.05 ^c^	0.03 ^a^	0.03 ^d^
Yazd (MS)	5.94 ^a^	133.58 ^e^	4.10 ^f^	45.00 ^i^	3.13 ^b^	1.80 ^b^	0.077 ^a b^	0.011 ^a^	0.24 ^a^	0.11 ^c^	0.023 ^c^	0.044 ^c^
Yazd (SS)	3.92 ^i^	77.32 ^l^	3.97 ^g^	30.74 ^n^	3.50 ^b^	1.60 ^c^	0.125 ^c^	0.008 ^c^	0.240 ^a^	0.137 ^b^	0.025 ^b^	0.072 ^b^

Means with a different letter are statistically significant at the 5% level of probability. C: Control; LS: Low stress; MS: Moderate stress; SS: Severe stress; QE: Quercetin equivalent; TAE: Tannic acid equivalent; TFC: Total flavonoid content; TPC: Total phenolic content; ESO: Essential oil yield; MAD: Malondialdehyde content; H_2_O_2_: Hydrogen peroxide content; GPX: Guaiacol peroxidase activity; APX: Ascorbate peroxidase activity; Pro: Protein; Chla: Chlorophyll a content; Chlb: Chlorophyll b content, Car: Carotenoid.

**Table 4 ijms-24-10438-t004:** Volatile compounds (%) of essential oils in studied ajowan populations under different salt concentrations and control conditions.

Population		Arak				Esfahfo				Qazvin				Yazd			
Compounds (%)	*RI^a^	C	LS	MS	SS	C	LS	MS	SS	C	LS	MS	SS	C	LS	MS	SS
α-Thujene	927	0.36	0.93	0.59	0.57	0.31	0.9	1.15	0.72	0.42	0.58	1.15	1.04	0.94	0.82	0.87	0.61
α-Pinene	941	0.15	0.27	0.16	0.17	0.11	0.23	0.30	0.22	0.13	0.16	0.31	0.27	0.23	0.21	0.21	0.17
Sabinene	972	0.18	0.38	0.24	0.22	0.21	0.35	0.48	0.31	0.25	0.21	0.68	0.42	0.58	0.40	0.44	0.24
β-Pinene	979	1.01	1.00	0.35	0.46	0.66	0.42	0.46	0.78	0.43	0.32	0.81	0.43	0.25	0.34	0.20	0.41
Myrcene	996	0.35	1.02	0.72	0.63	0.47	1.16	1.43	1.00	0.48	0.88	1.44	1.28	1.19	1.10	1.17	0.87
α-Terpinene	1017	0.38	0.75	0.53	0.66	0.19	0.74	0.85	0.67	0.34	0.67	0.94	0.84	0.77	0.74	0.82	0.62
*p*-Cymene	1025	18.74	22.18	25.42	22.34	20.32	26.16	25.71	19.21	18.80	25.67	23.09	25.10	24.06	21.73	20.55	22.8
β-Thujone	1110	0.81	1.11	0.79	0.95	0.74	0.18	0.24	0.34	0.77	1.04	1.14	1.13	1.06	0.88	0.96	1.00
γ-Terpinene	1057	26.89	21.94	26.71	24.27	23.1	22.75	25.51	21.71	22.85	32.81	23.02	30.83	31.82	25.55	22.00	24.33
Pulegone	1246	0	0	0	0	0.71	0.08	0.03	0.02	0	0	0	0	0.02	0.01	0.03	0.07
Terpinene-4-ol	1181	0.21	0.23	0.31	0.27	0.29	0.35	0.12	0.12	0.31	0.27	0.13	0.30	0.27	0.21	0.30	0.34
Thymol	1290	50.16	50.5	40.08	47.92	52.2	37.63	42.11	51.58	54.29	32.7	48.67	36.44	36.79	44.92	49.83	47.83
Carvacrol	1315	0.65	0.85	0.63	0.62	0.63	0.45	0.82	0.98	0.66	0.45	0.78	0.77	0.75	1.06	0.91	0.55
Total	-	99.80	98.75	98.94	99.08	99.99	98.19	92.42	97.66	99.46	99.36	98.56	98.85	98.73	98.15	98.11	99.84

*RI^a^: Calculated retention index. C: Control; LS: Low stress; MS: Moderate stress; SS: Severe stress conditions.

**Table 5 ijms-24-10438-t005:** Principal component loadings for the measured traits of the ajowan populations.

Seed Constitute	PC1	PC2
*p*-cymene	0.465	0.091
y-terpinene	0.537	−0.016
Thymol	−0.568	−0.087
Essential oil yield	0.204	0.451
Total phenolic content	−0.147	−0.004
Total flavonoid content	−0.295	0.368
DPPH radical scavenging	−0.0001	−0.640
Reducing power (500 mg/L)	−0.139	0.483
Eigenvalue	2.696	1.956
Percent of variance	33.706	24.455
Cumulative percentage	33.706	58.161

**Table 6 ijms-24-10438-t006:** Correlation coefficients between bioactive components from studied ajowan populations under control and low salt conditions (below diagonal) and under control and low salt stress conditions (on diagonal).

Traits								
*p*-Cymene	1	0.37 ^n.s^	−0.870 ^n.s^	−0.28 ^n.s^	−0.643 ^n.s^	−0869 ^n.s^	0.947 *	0.007 ^n.s^
γ-Terpinene	0.796 ^n.s^	1	−0.733 ^n.s^	−0.188 ^n.s^	0.210 ^n.s^	0.114 ^n.s^	0.390 ^n.s^	0.003 ^n.s^
Thymol	−0.93 ^n.s^	−0.961 *	1	0.076 ^n.s^	0.475 ^n.s^	0.580 ^n.s^	−0.773 ^n.s^	−0.225 ^n.s^
Essential oil yield	0.430 ^n.s^	0.804 ^n.s^	−0.669 ^n.s^	1	−0.429 ^n.s^	0.062 ^n.s^	−0.573 ^n.s^	0.953 ^n.s^
Total phenolic content	−0.642 ^n.s^	−0.56 ^n.s^	0.614 ^n.s^	−0.707 ^n.s^	1	0.874 ^n.s^	−0.401 ^n.s^	−0.597 ^n.s^
Total flavonoid content	−0.524 ^n.s^	−0.219 ^n.s^	0.383 ^n.s^	0.403 ^n.s^	−0.289 ^n.s^	1	−0.758 ^n.s^	−0.152 ^n.s^
DPPH ^a^	−0.780 ^n.s^	−0.990 **	0.919 ^n.s^	−0.872 ^n.s^	0.574 ^n.s^	0.093 ^n.s^	1	−0.309 ^n.s^
Reducing power	−0.165 ^n.s^	−0.712 ^n.s^	0.499 ^n.s^	−0.914 ^n.s^	0.361 ^n.s^	−0.407 ^n.s^	0.799 ^n.s^	1

^n.s^: non-significant. ** Significant at 1% level of probability. * Significant at 5% level of probability. ^a^ 1,1-Diphenyl-2 picrylhydrasyl.

**Table 7 ijms-24-10438-t007:** Correlation coefficients between bioactive components from studied ajowan populations under moderate and serve salt conditions (below diagonal) and under moderate salt stress and severe stress conditions (on diagonal).

Traits								
*p*-Cymene	1	0.944 *	−0.927 ^n.s^	0.969 ^n.s^	−0.268 ^n.s^	−0.949 ^n.s^	−0.088 ^n.s^	−0.270 ^n.s^
γ-Terpinene	0.931 ^n.s^	1	−0.998 ^n.s^	0.905 ^n.s^	0.032 ^n.s^	−0.888 ^n.s^	0.221 ^n.s^	−0.388 ^n.s^
Thymol	−0.916 ^n.s^	−0.994 **	1	−0.887 ^n.s^	−0.079 ^n.s^	0.869 ^n.s^	−0.267 ^n.s^	0.405 ^n.s^
Essential oil yield	−0.461 ^n.s^	−0.154 ^n.s^	0.179 ^n.s^	1	−0.187 ^n.s^	−0.843 ^n.s^	−0.028 ^n.s^	−0.463 ^n.s^
Total phenolic content	−0.815 ^n.s^	−0.939 ^n.s^	0.970 *	0.138 ^n.s^	1	0.404 ^n.s^	0.980 ^n.s^	−0.637 ^n.s^
Total flavonoid content	−0.472 ^n.s^	−0.359 ^n.s^	0.257 ^n.s^	−0.006 ^n.s^	0.019 ^n.s^	1	0.219 ^n.s^	−0.026 ^n.s^
DPPH ^a^	0.584 ^n.s^	0.265 ^n.s^	−0.211 ^n.s^	−0.732 ^n.s^	−0.017 ^n.s^	−0.668 ^n.s^	1	−0.663 ^n.s^
Reducing power	−0.549 ^n.s^	−0.516 ^n.s^	0.419 ^n.s^	−0.142 ^n.s^	0.204 ^n.s^	0.969 *	−0.527 ^n.s^	1

^n.s^: non-significant. ** Significant at 1% level of probability. * Significant at 5% level of probability. ^a^ 1,1-Diphenyl-2 picrylhydrasyl.

**Table 8 ijms-24-10438-t008:** Analysis of variance for the measured traits from studied ajowan populations.

Sources of Variation	df	DPPH	RP	TFC	TPC	NCP	ESO	Thymol	γ-Terpinene	*p*-Cymene
Gen	3	3,499,789 **	0.115 **	1.03 **	2882.8 **	663.74 **	3.27 **	39.73 **	40.08 **	3.05 **
Salt	3	27,167,123 **	0.323 **	3.51 **	8003.4 **	919.11 **	0.44 **	97.89 **	10.20 **	30.41 **
Gen × Salt	9	66623 **	0.015 **	0.82 **	2543.8 **	344.43 **	1.27 **	181.4 **	47.95 **	22.06 **
Error	32	1875.01	0.001	0.002	0.018	128.42	0.033	0.081	0.181	0.006
Coeff Var	-	1.16	13.37	1.21	0.11	24.37	4.91	0.63	1.68	0.35
**Sources of Variation**	**df**	**MAD**	**H_2_O_2_**	**GPX**	**APX**	**Pro**	**Chla**	**Chlb**	**Car**	
Gen	3	1.24 **	1.73 **	0.008 **	0.0002 **	0.00003 **	0.01 **	0.001 **	0.002 **	
Salt	3	0.97 **	0.22 **	0.031 **	0.00008 **	0.00004 **	0.01 **	0.001 **	0.001 **	
Gen × Salt	9	5.99 **	0.49 **	0.024 **	0.0001 **	0.00001 **	0.028 **	0.001 **	0.005 **	
Error	32	0.056	0.029	0.0002	0.0000004	0.00002	0.0004	0.0003	0.0006	
Coeff Var	-	8.05	10.26	11.81	5.19	2.04	11.47	32.08	8.24	

** indicate significant differences at *p* < 0.05. TFC: Total flavonoid content; TPC: Total phenolic content; ESO: Essential oil yield; MAD: Malondialdehyde content; H_2_O_2_: Hydrogen peroxide content; GPX: Guaiacol peroxidase activity; APX: Ascorbate peroxidase activity; Pro: Protein; Chla: Chlorophyll a content; Chlb: Chlorophyll b content, Car: Carotenoid.

**Table 9 ijms-24-10438-t009:** Two-way multivariate analysis of variance by three specified methods.

Sources of Variation	df	Value
Gen		
Wilks’ Lambda	51	0.0000006 **
Pillai’s Trace	51	2.9457172 **
Hotelling-Lawley Trace	51	1133.2316064 **
Stress		
Wilks’ Lambda	51	0.00000650 **
Pillai’s Trace	51	2.88597805 **
Hotelling-Lawley Trace	51	380.54326555 **

** indicates significant differences at *p* < 0.01.

**Table 10 ijms-24-10438-t010:** Morphological traits in open- and self-pollinated populations of ajowan.

Accession Code	No. of Umbels			No. of Umbellules per Inflorescence			No. of Flowers per Umbels		
	S	OP	ID	S	OP	ID	S	OP	ID
Esfahfo	71.67	58.33	13.33	12.33	12.67	−0.33	163	140.67	22.33
Yazd	69	86	−17	18.33	15	3.33	222.33	213.67	8.67
Qazvin	65.33	40.33	25	17.67	13.33	4.33	229.33	192	37.33
Arak	52.67	153.33	−100.67	16	16.67	−0.67	242.67	228.67	14
**Accession Code**	**Height (cm)**			**No. of Flowering Branches**			**Inflorescence Diameter (cm)**		
	**S**	**OP**	**ID**	**S**	**OP**	**ID**	**S**	**OP**	**ID**
Esfahfo	109.67	117.67	−8	9.33	8	1.33	2.33	2.83	−0.5
Yazd	107	107	0	7.33	6.67	0.67	2.67	2.67	0
Qazvin	122.67	100	22.67	12.67	8	4.67	3.17	2.5	0.67
Arak	104.33	112	−7.67	10.67	22	−11.33	2.77	3.83	−1.07
**Accession Code**	**No. of Seeds per Umbel**			**Essential Oil Yield (%)**			**Crown Cover Diameter (cm^2^)**		
	**S**	**OP**	**ID**	**S**	**OP**	**ID**	**S**	**OP**	**ID**
Esfahfo	326	281.33	44.67	1.5	5.32	−3.82	596.6	1227.22	−630.62
Yazd	444.67	427.33	17.33	4.54	5.51	−0.96	558.4	429.66	128.74
Qazvin	458.67	384	74.67	2.07	4.73	−2.65	1452.77	727.96	724.82
Arak	485.33	457.33	28	3.01	5.08	−2.07	928.92	858.27	70.65

**Table 11 ijms-24-10438-t011:** Geographical location of 28 ajowan populations.

No	Accession Number	Location	Accession Code	Geographical Region	Latitude	Longitude	Altitude (m)
1	37,477	Nahadjan, Khorasan, Iran	Khorsar	East	32°32′ N	59°47′ E	1816
2	38,913	Boshruieh, Khorasan, Iran	Khorbos	East	33°53′ N	57°27′ E	880
3	38,924	Birjand, Khorasan, Iran	Khorbir	East	32°53′ N	59°13′ E	1461
4	38,929	Sarbisheh, Khorasan, Iran	Khorsa	East	32°34′ N	59°48′ E	1827
5	37,492	Boztanj, Khorasan, Iran	Khorbi	East	32°51′ N	59°12′ E	1458
6	37,483	Mohammadieh, Khorasan, Iran	Khormo	East	32°55′ N	59°13′ E	1460
7	37,529	Ghayen, Khorasan, Iran	Khorgh	East	33°43′ N	59°10′ E	1455
8	15,226	Khomein, Markazi, Iran	Arakkho	West	33°38′ N	50°4′ E	1811
9	14,743	Arak, Markazi, Iran	Arak	West	34°5′ N	49°42′ E	1735
10	14,492	Shazand, Markazi, Iran	Araksha	West	33°56′ N	49°24′ E	1914
11	14,322	Hamedan, Hamedan, Iran	Hamdan	West	34°47′ N	48°30′ E	1818
12	37,251	Mollasadra, Yazd, Iran	Yazmol	Center	31°50′ N	54°22′ E	1242
13	31,831	Markaztahghighat, Yazd Iran	Yaztah	Center	31°54′ N	54°16′ E	1213
14	33,683	Saduq, Yazd, Iran	Yazsad	Center	32°1′ N	53°28′ E	2091
15	15484	Shahedieh, Yazd, Iran	Yazshah	Center	31°56′ N	54°16′ E	1193
16	15,864	Sadooqi, Yazd, Iran	Yazist	Center	31°52′ N	54°20′ E	1228
17	1085	Yazd, Yazd, Iran	Yazd	Center	31°53′ N	54°21′ E	1215
18	4077	Ghahderijan, Isfahan, Iran	Esfahgh	Center	32°34′ N	51°26′ E	1615
19	943	Fozveh, Isfahan, Iran	Esfahfo	Center	32°36′ N	51°26′ E	1615
20	20,055	Qazvin, Qazvin, Iran	Qazvin	North	36°16′ N	49°59′ E	1305
21	906	Tehran, Tehran, Iran	Tehran	North	35°41′ N	51°23′ E	1168
22	10,569	Ardabil, Ardabil, Iran	Ardebil	Northwest	38°16′ N	48°18′ E	1332
23	17,902	Marvdasht, Fars, Iran	Farsmar	South	29°52′ N	52°49′ E	1600
24	17,861	Shiraz, Fars, Iran	Farsfars	South	29°35′ N	52°35′ E	1508
25	23,011	Rafsanjan, Kerman, Iran	Rafsanj	South	30°21′ N	56°0′ E	1545
26	-	Genebank IPK	IPK1	-	-	-	-
27	-	Genebank IPK	IPK2	-	-	-	-
28	-	Genebank IPK	IPK3	-	-	-	-

## Data Availability

The data will be available based on request.

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
