# Peer review of "Phytochemical, Morphological, and Physiological Variation in Different Ajowan (*Trachyspermum ammi* L.) Populations as Affected by Salt Stress, Genotype × Year Interaction and Pollination System"

_ijms, 2023, doi:10.3390/ijms241310438_

Round 1

Reviewer 1 Report

Dear Authors,

Congratulations on your manuscript, which I read with pleasure. The manuscript aims (1) to evaluate the morphological, oil, and yield-related traits in 28 ajowan populations grown in two consecutive years, (2) to 82 evaluate the effects of salt stress on physiological traits, seed yield and related traits, essential oil composition, polyphenolic contents, and antioxidant capacity of selected ajowan populations, (3) to assess the effect of self-pollination on some important morphological traits. The topic is of interest, as the authors highlight, due to the relevance of ajowan for medicinal, food, and industrial proposes. Writing is in general very efficient, but sometimes I have the feeling that the results section is a bit descriptive. In my opinion, it would be better if the results and discussion were two different sections. The discussion could also be improved if the way how these sections are presented is changed.

Conclusions are in general supported by the results. I found some minor typo errors during the reading of the MS which I advise the authors to implement before publication. Statistical analysis description, as well as figures and tables caption, also need to be improved. Please consider the comments below. 

Introduction

Line 55: “but it is also valuable sources of amino acids…” replace by “it is also a valuable source of”

Line 63-64: Please rewrite.

Materials and Methods

2.1 Section:

-what about the environmental conditions? Was it done during springer, summer? At least you should give the months and year of the trial so that the reader can have a perception of the environmental conditions. How long did the trail last?

-Where is Fig S1a?

2.2 Section:

-Where is Fig S1b?

2.5 Section:

- Line 142: please wright total flavonoid content before TPC, as this is the first time you mention it

2.6.1. Please specify how you did the calculations based on IC50

2.9 Section:

Why only measure APX activity and not also SOD and CAT, for instance? Is there any valid reason to measure only APX? Please specify it

2.11 Section:

The description is incomplete. How did you estimate it? Spectrophotometrically? If yes which wavelength was used?

2.13 Section:

Please specify how long did the trial last.

2.14 Section:

Please specify the specific statistical analysis that you perform. I.e. the specific analysis of variance and the respective post-hoc test for the different experiments that you performed

Results and Discussion

Lines 218, 239, l425-  This lines clearly show that the environment is playing a role (obviously, and as you also mention) in the oil content of the population. This is why it is really important to clearly specify the growth (environmental) conditions in the two different years, as well as the months/years where the trial occurred in the M&M section.

Table 3 – It would be essential to present also the SD or SEM means. You should also clearly specify the replicates number in the figures caption. This is valid for all your figures/tables

Figure 3 – you should specify if the bars correspond to the SD or the SEM. Also, it is a bit strange that no variation can be seen in these bars.

Table 7 and 8. Please specify in both table captions that n.s = non-significant.

Author Response

Reviewer 1.

Dear Authors,

Congratulations on your manuscript, which I read with pleasure. The manuscript aims (1) to evaluate the morphological, oil, and yield-related traits in 28 ajowan populations grown in two consecutive years, (2) to  evaluate the effects of salt stress on physiological traits, seed yield and related traits, essential oil composition, polyphenolic contents, and antioxidant capacity of selected ajowan populations, (3) to assess the effect of self-pollination on some important morphological traits. The topic is of interest, as the authors highlight, due to the relevance of ajowan for medicinal, food, and industrial proposes. Writing is in general very efficient, but sometimes I have the feeling that the results section is a bit descriptive. In my opinion, it would be better if the results and discussion were two different sections. The discussion could also be improved if the way how these sections are presented is changed.

Conclusions are in general supported by the results. I found some minor typo errors during the reading of the MS which I advise the authors to implement before publication. Statistical analysis description, as well as figures and tables caption, also need to be improved. Please consider the comments below. 

Answer: I firstly appreciate the respected reviewer for his/her time for valuable comments. We did all comments according to suggestions.

Introduction

Line 55: “but it is also valuable sources of amino acids…” replace by “it is also a valuable source of”

Done.

Line 63-64: Please rewrite.

This sentence has been removed. 

Materials and Methods

2.1 Section:

What about the environmental conditions? Was it done during springer, summer? At least you should give the months and year of the trial so that the reader can have a perception of the environmental conditions. How long did the trail last?

Added to the article.

Where is Fig S1a?

This figure is in the supplementary file.

2.2 Section:

Where is Fig S1b?

This figure is in the supplementary file.

2.5 Section:

Line 142: please wright total flavonoid content before TPC, as this is the first time you mention it.

Done.

2.6.1. Please specify how you did the calculations based on IC50.

Done.

2.9 Section:

Why only measure APX activity and not also SOD and CAT, for instance? Is there any valid reason to measure only APX? Please specify it.

The results of salinity stress of APX and GPX enzymes were  significant and used in the article.

2.11 Section:

The description is incomplete. How did you estimate it? Spectrophotometrically? If yes which wavelength was used?

The information were Completed.

2.13 Section:

Please specify how long did the trial last.

It is mentioned in the article.

2.14 Section:

Please specify the specific statistical analysis that you perform. I.e. the specific analysis of variance and the respective post-hoc test for the different experiments that you performed.

Done.

Results and Discussion

Lines 218, 239, l425-  This lines clearly show that the environment is playing a role (obviously, and as you also mention) in the oil content of the population. This is why it is really important to clearly specify the growth (environmental) conditions in the two different years, as well as the months/years where the trial occurred in the M&M section.

The detailed information added according to suggestion. The environment has a significant effect on increasing the performance of the essential oil. In this article, the effect of salinity stress on ajowan population has been studied and the conditions of salinity stress have been explained.

Table 3 – It would be essential to present also the SD or SEM means. You should also clearly specify the replicates number in the figures caption. This is valid for all your figures/tables.

In the materials and methods section, it is mentioned that all experiments were done with three repetitions, and in table 3, LSD numbers are expressed, and for this reason, SE are not stated.

Figure 3 – you should specify if the bars correspond to the SD or the SEM. Also, it is a bit strange that no variation can be seen in these bars.

All the bars are drawn with SE, and the vertical axis of numbers is with a scale of 1000 µg/mL, so the difference between the bars is not clear, and if you want, bars are removed from the chart and expressed only significant letters.

Table 7 and 8. Please specify in both table captions that n.s = non-significant.

Done.

Reviewer 2 Report

The authors of the manuscript titled “Variation in chemical composition and morpho-physiological traits in different Ajowan (Trachyspermum ammi L.) populations as affected by salinity, genotype × year interaction and pollination system” The current experiment was conducted (1) to evaluate the morphological, oil, and yield-related traits in ajowan populations grown in two consecutive years, (2) to evaluate the effects of salt stress on physiological traits, seed yield and related traits, essential oil composition, polyphenolic contents, and antioxidant capacity of selected ajowan populations, (3) to assess the effect of self-pollination on some important morphological traits and (4) to use multivariate analyses for better classification and interpretation of data.

General comments

Overall, the study is well-designed and presented in a good way.

Abstract

The authors are requested to clarify the last sentence from lines 29-31.

Introduction

This section is also well written but the authors are requested to add recent references in this section.

Materials and Methods

The authors are requested to add the relevant references to all sections of the Materials and Methods.

Results

This section of the manuscript is presented well but the authors are requested to separate the result section from the Discussion section of the manuscript.

Discussion

The authors are requested to separate the Discussion section from the Result section of the manuscript. Furthermore, the authors are requested to use more relevant and recent references to support the outcomes of their current work.

Conclusion

The authors are requested to rewrite sentences from lines 627-628 to avoid repetition of words and make the sentence clear.

Furthermore, the authors are requested to present the studied species in ascending order in terms of the studied EO yields etc. For example, they can arrange the studied plants in this sequence 

Arak > Yazd > Yazd > and so on..

Author Response

Reviewer2.

The authors of the manuscript titled “Variation in chemical composition and morpho-physiological traits in different Ajowan (Trachyspermum ammi L.) populations as affected by salinity, genotype × year interaction and pollination system” The current experiment was conducted (1) to evaluate the morphological, oil, and yield-related traits in ajowan populations grown in two consecutive years, (2) to evaluate the effects of salt stress on physiological traits, seed yield and related traits, essential oil composition, polyphenolic contents, and antioxidant capacity of selected ajowan populations, (3) to assess the effect of self-pollination on some important morphological traits and (4) to use multivariate analyses for better classification and interpretation of data.

General comments

Overall, the study is well-designed and presented in a good way.

Answer: I firstly appreciate the respected reviewer for his/her time for valuable comments. We did all comments according to suggestions.

Abstract

The authors are requested to clarify the last sentence from lines 29-31.

Done.

Introduction

This section is also well written but the authors are requested to add recent references in this section.

Done.

Materials and Methods

The authors are requested to add the relevant references to all sections of the Materials and Methods.

Done.

Results

This section of the manuscript is presented well but the authors are requested to separate the result section from the Discussion section of the manuscript.

The correction was made based on the suggestion.

Discussion

The authors are requested to separate the Discussion section from the Result section of the manuscript. Furthermore, the authors are requested to use more relevant and recent references to support the outcomes of their current work.

New and related references were added to the article.

Conclusion

The authors are requested to rewrite sentences from lines 627-628 to avoid repetition of words and make the sentence clear.

Furthermore, the authors are requested to present the studied species in ascending order in terms of the studied EO yields etc. For example, they can arrange the studied plants in this sequence 

Arak > Yazd > Yazd > and so on.

Done.

Round 2

Reviewer 2 Report

I accept this manuscript because the authors answered my all queries very well.